# Dietary Exposure Assessment of Veterinary Antibiotics in Pork Meat on Children and Adolescents in Cyprus

**DOI:** 10.3390/foods9101479

**Published:** 2020-10-16

**Authors:** Demetra Kyriakides, Andreas C. Lazaris, Konstantinos Arsenoglou, Maria Emmanouil, Olympia Kyriakides, Nikolaos Kavantzas, Irene Panderi

**Affiliations:** 1Laboratory of Pathological Anatomy, Department of Clinical and Laboratory Medicine, School of Medicine, National and Kapodistrian University of Athens, 75, Mikras Asias Avenue, Goudi, 11527 Athens, Greece; alazaris@med.uoa.gr (A.C.L.); nkavantz@med.uoa.gr (N.K.); 2Veterinary Services, Ministry of Agriculture, Rural Development and Environment, 1417 Nicosia, Cyprus; carsenoglou@vs.moa.gov.cy (K.A.); memmanouil@vs.moa.gov.cy (M.E.); 3Archbishop Makarios III Hospital, 2012 Nicosia, Cyprus; olympiaieronym@gmail.com; 4Laboratory of Pharmaceutical Analysis, Panepistimiopolis, Division of Pharmaceutical Chemistry, Department of Pharmacy, National and Kapodistrian University of Athens, Zografou, 15771 Athens, Greece

**Keywords:** food composition, veterinary antibiotics, dietary exposure assessment, child, adolescent, pork meat, Cyprus

## Abstract

In recent years, huge amounts of antibiotics have been administered to farm animals, and as a result, residues of these antibiotics can accumulate in livestock products and, once consumed, may be transmitted to humans. Farm animals’ antibiotic treatment may therefore present a risk for consumers health, especially for children and adolescents. In children, the immune system is not fully developed, and thus, they are more susceptible than adults to resistant bacteria. A dietary exposure assessment was conducted on veterinary antibiotics found in raw pork meat among children and adolescents in Cyprus, since pork is the most consumed red meat in Cypriot population. The study was based on the results of the occurrence of 45 residual antibiotics in raw pork meat samples in Cyprus between 2012 and 2017 in combination with data on the consumption of pork meat on children and adolescents taken from the latest demographic report in Cyprus. Estimated daily intake (EDI) values of veterinary antibiotics for children aged 6–9 years old, were higher compared to EDI values for adolescents aged 10–17 years old. The percentage ratio of the estimated daily intake to the acceptable daily intake for all the veterinary antibiotic residues was less than 5.6. The results indicate that antibiotic residues in pork meat of inland production are below the acceptable daily intake and are of low risk to human health related to the exposure of antibiotics. Nevertheless, continuous exposure to low levels of antibiotic residues in respect to age vulnerability should be of a great concern.

## 1. Introduction

The modern fight against bacterial infections dates back to the antibiotic era in the middle 1900s and to the discovery of the first antibiotics [1]. Since then, antibiotics have constantly been used not only for the treatment of various infections in humans but also in food production [2]. Huge amounts of antibiotics are constantly used as growth promoters or for prophylaxis and the treatment of infections among farm animals [3]. As a result, residues of these antibiotics can accumulate in livestock products and, once consumed, may then be transmitted to humans [4,5]. Exposure in low antibiotics concentrations may cause several adverse effects to human health, including toxicity, hypersensitivity, neurological disorders, gastrointestinal disturbances, and multidrug resistance and may also increase the prevalence of antibiotic resistance genes [6]. Antimicrobial resistance is a complicated phenomenon involving many interconnected factors between humans, animals, and environmental sources [7]. Multidrug-resistant bacteria are currently considered as an emerging global disease and a serious health concern [8]. In fact, antimicrobial resistance has reached alarming levels worldwide, and as a result, many of the treatments for common infections tend to be ineffective [9]. However, there is an insufficient understanding about whether or how human exposure to low concentrations of veterinary antibiotics in food affects the intestinal microflora. In children, the exposure to antibiotics is related to the development of bacterial resistance, childhood obesity and diabetes mellitus along with disturbances of glucose homeostasis [10,11,12,13,14]. The pharmacokinetics and pharmacodynamics of various drugs in children may differ significantly in relation to adults [15,16]. Certain antibiotics such as sulfonamides, chloramphenicol, and tetracyclines due to the known toxicity should not be used in neonates [17]. 

Effective national food control systems are crucial for the protection of the health and safety of consumers. Currently, the use of antibiotics in farm animals has been regulated worldwide, and maximum residue limits (MRLs) have been established for veterinary antibiotics in foodstuffs of animal origin [18,19,20,21]. Despite the extensive use of antibiotics in animals, reliable data about the pattern of use and the quantity of these compounds are not always available [22]. That is why it is important to monitor the concentration levels of antibiotic residues in animals and in food products of animal origin [23,24,25,26,27,28,29,30,31]. Studies on the quantitative risk assessment of foods provide important frameworks to address some of the issues related to the risk to human health from the intake of various residues via food [32,33,34,35,36]. Recently, a study on the residue concentrations and exposure levels of ethoxyquin and ethoxyquin dimer in farmed aquatic animals in Korea has been published [37]. A dietary exposure assessment study is the first step of a risk assessment procedure.

The MRL values for veterinary antibiotics in porcine tissue and pork meat as established under the European Regulation [18] and the Codex Alimentarius [19,20] must be controlled by the national veterinary authorities. Nevertheless, antibiotic residues in pork meat at concentrations above the established MRLs are still reported [38]. Pork is the most consumed red meat in Cyprus, but pig antibiotic treatment may present a risk for consumer health, especially for children and adolescents. In children the immune system is not fully developed, and thus, they are more susceptible than adults to resistant bacteria [11,39]. The risk of consuming pork meat, especially among children and adolescents, is of concern due to lack of data for the long-term bioaccumulation of these compounds. The bioaccumulation of antibiotics in humans may cause hypersensitivity reactions, gastrointestinal disorders, liver damages, mutagenicity, carcinogenicity, and reproductive toxicity [40,41]. Several studies revealed that children are widely exposed to antibiotics from dietary intake at low dosage levels [42,43,44]. In addition, studies in mice revealed that exposure in low concentrations of residual antibiotics during developmental periods affects the metabolism, gut microbiota, and adipogenesis, which can lead to obesity and diabetes [45,46].

In light of the above, we thought that it would be of interest to conduct a dietary exposure assessment that could be used as the preliminary step of a risk assessment procedure over the consumption of pork meat among children and adolescents in the Cypriot population. When including a dietary exposure assessment, the risk assessment provides the scientific basis for the establishment of appropriate regulations and guidelines on the use of veterinary antibiotics in food producing animals. This will ensure that safety requirements will be protective for the consumers and appropriate for use in national and international scale. To this purpose, the results of the occurrence of 45 residual antibiotics in pork meat samples between 2012 and 2017, presented in our previous work by Kyriakides et al. [47], were combined with food consumption data of raw pork meat among children and adolescents in Cyprus. This study aims to clarify that a systematic exposure of antibiotics in low concentrations in early life may have negative impact in human health. At the same time, a comparison analysis is being developed in comparison with adults, since children have higher exposure levels per kg of body weight and thus are more susceptible to various residues. Based on the above the presented dietary exposure assessment to antibiotics for young population between 6 and 17 years of age is important and could help in the design of safety measures for the protection of the young population. This research proved that the antibiotic residues in pork meat of inland production are below the maximum established acceptable daily intake (ADI) and compose low direct risk to human health. Nevertheless, continuous exposure to low levels of antibiotics in respect to age vulnerability should be of a great concern, and further studies are needed.

## 2. Materials and Methods

### 2.1. Pork Meat (Muscle Tissue) Sample Collection

For six continuously years (2012–2017), among all districts of the Republic of Cyprus, 15,484 pork meat samples were collected from authorized slaughterhouses. The raw pork meat samples were collected from authorized slaughterhouses situated among the five districts of the Republic of Cyprus: Famagusta, Limassol, Nicosia, Larnaka, and Paphos. These samples were carried in polypropylene bags placed in icebox and stored at −18 °C before analysis.

### 2.2. Chemicals and Reagents

All solvents were of High-Performance Liquid Chromatography (HPLC) grade and purchased from E. Merck (Darmstadt, Germany). Reference standards of the veterinary antimicrobials and deuterated internal standards were of pharmaceutical purity grade and purchased from Sigma-Aldrich (St. Louis, MO, USA).

### 2.3. Microbial Screening Method for the Detection of Antibiotic Residues in Slaughter Animals

A microbial inhibitions assay, Premi^®^Test purchased from R-Biopharm AG (Darmstadt, Germany) was used to screen collected meat samples, since it detects a broad spectrum of the most used veterinary antibiotics in animal husbandry [48]. The Premi^®^Test is based on the inhibition of the growth of *Bacillus stearothermophilus* in the presence of antibiotic residues.

### 2.4. Liquid Chromatographic-Tandem Mass Spectrometric Analysis

Pork tissue samples were analyzed by a specific liquid chromatography tandem spectrometry method that was validated to detect and quantitate 45 commonly used veterinary antibiotics as described by Kyriakides et al. [47]. The confirmatory LC-MS/MS method was conducted on a Waters Alliance 2695/Quattro Premier^TM^ XE mass spectrometer. The sample preparation involved homogenization, addition of a mixed solution of deuterated internal standards, and extraction with 5% trichloroacetic acid in water for aminoglycosides, fluoroquinolones, quinolones, and tetracyclines or extraction with 0.2 M ammonium acetate solution in acetonitrile for β-lactam’s, sulfonamides, macrolides, dehydroreductase inhibitor, and pleuromutilin. Chromatographic separation of tetracyclines, quinolones and aminoglycosides was carried out using a Luna C18 HST analytical column (100 × 20 mm, 2.5 μm particle size), while for the analysis of sulfonamides, β-lactams, macrolides, trimethoprim, and tiamulin, a Symmetry C18 Waters analytical column (150 × 2.1 mm, 3.5 μm particle size) was used. Data acquisition and analysis were performed using MassLynx™ Waters Software, Waters Corporation (Milford, MA, USA) and the OpenLynx™ Waters software Waters Corporation (Milford, MA, USA) was used for the screening, identification, and quantitation of the positive results. Data on the MRL values of these veterinary antibiotics including six aminoglycosides, three cephalosporins, seven fluoroquinolones, one quinolone, one lincosamide, three macrolides, five penicillin’s, two β-lactam’s, 11 sulphonamides, one dehydroreductase (DHR) inhibitor, and four tetracyclines, are presented in Table 1.

## 3. Results and Discussion

### 3.1. Screening and Confirmatory Results

A microbial screening method was used for the determination of antibiotic residues in 15,484 raw pork meat samples. A total of 1766 samples suspected positive samples were analyzed by means of a tandem LC-MS/MS confirmatory method which was presented in detail in our previously published work [47]. From this research, 13 veterinary antibiotic residues including one aminoglycoside, one fluoroquinolone, one lincosamide, one macrolide, one β-lactam, three sulfonamides, one dehydroreductase inhibitor, and four tetracyclines were detected in 596 positive samples. In 74.6% of the positive samples (445 samples), all the 45 tested antibiotic residues were below MRLs. In 25.3% of the positive samples (151 samples), nine antibiotics were detected at concentration levels above the MRLs.

### 3.2. Daily Consumption of Raw Pork Meat for Children and Adolescent

The total Daily Consumption (DC_T_) of raw pork meat per person per family, was computed according to the following equation:(1)$${\mathrm{DC}}_{T}=\left(\frac{\frac{\left(\left(\mathrm{annual}\;\mathrm{family}\;\mathrm{expenses}\;\mathrm{for}\;\mathrm{pork}\;\mathrm{meat})/(\mathrm{retail}\;\mathrm{selling}\;\mathrm{price}\;\right)\right)\times 1000}{365}}{\mathrm{average}\;\mathrm{number}\;\mathrm{ofpersons}\;\mathrm{per}\;\mathrm{family}}\right)$$

The annual family expenses for pork meat was estimated based on the statistical data of a survey conducted in 2017 by the Statistical Service of Cyprus [49]. Based on this data, the retail selling price for pork meat was €4.8/kg, and the average number of persons per family was 2.73 ± 0.04. Based on the above, the total daily consumption (DC_T_) of raw pork meat per person per family was estimated at 58.13 g of pork meat per person per day.

### 3.3. Dietary Exposure Assessment of Veterinary Antibiotics in Children and Adolescents

The current dietary exposure assessment was based on the results about the occurrence of 45 residual antibiotics in pork meat samples in Cyprus between 2012 and 2017 [47]. The lower-bound (LB) approach was used for the left censored data of the microbial screening method and the results below the limit of detection (LOD) were replaced by zero [50]. All the pork meat samples with antibiotics at concentrations above the established maximum residue limits (MRLs) were excluded from the exposure assessment since this pork meat is banned from the Cyprus market. In particular, the Veterinary Services of the Ministry of Agriculture, Rural Development, and Environment in Cyprus have the legal power to take appropriate measures in cases of detected residues violations, restriction or prohibition of the placing on the market; monitoring, recall, withdrawal, and/or destruction. A national program for testing of slaughtered pigs under which every animal is tested for the presence of antimicrobial residues. Animals are detained pending a result, if non-compliant, the animals are destroyed and in no case possible does the products reach the Cyprus market. Amoxycillin was therefore excluded from this study since it was detected during 2015 at a concentration range of 73.7–547 μg kg^−1^ and during 2016 at a concentration of 121 μg kg^−1^, which are above the established MRL limit in porcine muscle of 50 μg kg^−1^. All the analytes that were detected only once in a raw pork meat sample were also excluded. Therefore, dihydrostreptomycin and flumequine were detected only once during 2012 and 2013, respectively and they were excluded from this dietary exposure assessment. Sulfamethoxazole was detected during 2012 at a concentration range of 33.2–536 μg kg^−1^, during 2014 at a concentration range of 1.45–69.8 μg kg^−1^, and during 2015 at a concentration range of 1.45–51.8 μg kg^−1^. However, the acceptable daily intake (ADI) for sulfamethoxazole is not defined, and the EDI to ADI ratio was not calculated for this compound.

The estimated daily intake (EDI) of the veterinary antibiotic residues expressed in μg/kg bw/day was calculated based on the following equation:(2)$$\mathrm{EDI}=\;\frac{{\mathrm{DC}}_{T}\;\times 10^{-3}\times C}{{\mathrm{Average}\;\mathrm{body}\;\mathrm{weight}\;}_{\mathrm{per}\;\mathrm{age}\;\mathrm{and}\;\mathrm{gender}\;\mathrm{group}}}$$
where DC_T_ is the daily consumption of pork meat in g per person per day, C is the geometric mean of the concentrations of antibiotics in pork meat expressed in μg/kg, and the average body weight is expressed in kg. The EDI values were estimated for three age groups in both genders, the first ranged between 6 and 9 years old, the second between 10 and 13 years old, and the last between 14 and 17 years old; additional information on the percentage EDI to ADI ratio for adults 18+ years was also included in this study. Average body weight data on different age/gender groups in Cyprus have been used for this exposure assessment [51]. The average body weight for children aged 6 to 9 years old is 28.5 kg and 27.6 kg for males and females, respectively. The average body weight for adolescents aged 10 to 13 years old is 45.5 kg and 44.5 kg for males and females, respectively, and for adolescents aged 14 to 17 years old is 65.6 kg and 54.5 kg for males and females, respectively. The average weight of adults 18+ in Cyprus is 80.9 kg and 64.6 kg for males and females, respectively.

The geometric mean of the concentrations of antibiotics was calculated for six consecutive years, and the estimated daily intake values in μg/kg bw/day of antibiotic residues in the male and female young population in Cyprus between 2012 and 2017 are presented in Table 2 and Table 3, respectively.

For the dietary exposure assessment, the maximum geometric mean concentration of each residue was used to calculate the representative EDI values presented in Table 4 and Table 5 for males and females, respectively, as a more representative worst-case scenario.

For both males and females in all age groups, the highest EDI values were observed in 2013 for lincomycin, which ranged from 0.13 μg/kg bw/day to 0.30 μg/kg bw/day and 0.16 μg/kg bw/day to 0.32 μg/kg bw/day, respectively. Chlortetracycline, and doxycycline exhibited the second and third higher EDI values, respectively. The EDI value for chlortetracycline ranged from 0.08 μg/kg bw/day to 0.17 μg/kg bw/day in males and 0.09 μg/kg bw/day to 0.18 μg/kg bw/day in females. For doxycycline, the EDI value ranged from 0.07 μg/kg bw/day to 0.16 μg/kg bw/day in males and 0.09 μg/kg bw/day to 0.17 μg/kg bw/day in females. The EDI values for all the antibiotic residues were lower in males than in females and were ranged from 0.01 μg/kg bw/day to 0.30 μg/kg bw/day for males and 0.01 μg/kg bw/day to 0.32 μg/kg bw/day for females. EDI values were higher in children aged 6 to 9 years old than in adolescents of both groups, and these results agree with previous studies [52,53].

The EDI values and the acceptable daily intake (ADI) values of the antibiotic residues were used to calculate the percentage EDI to ADI ratio according to the Equation (3). Each ADI value is the maximum amount of a substance that a person can consume daily throughout his/her life without posing this any risk. An EDI to ADI ratio of less than unit means that the estimated daily intake of antibiotics is lower than the acceptable daily intake, i.e., no risk.
(3)$$\%\;\mathrm{EDI}\;\mathrm{to}\;\mathrm{ADI}\;\mathrm{ratio}=\frac{\mathrm{EDI}\;}{\mathrm{ADI}\;}\;\times 100$$

The results of the % EDI to ADI ratio are presented in Table 4 and Table 5 for male and female children, respectively. Based on the analytical data of our previous work on the detection of antibiotic residues in porcine meat samples [47], we have also calculated the percentage EDI to ADI ratio for adult men and women. The plots of the % EDI to ADI ratio versus the antibiotics for all groups including adults are presented in Figure 1 and Figure 2.

As it can be observed in Figure 1 and Figure 2, for both male and female adolescents in all age groups, the % EDI to ADI ratio was less than 5.6 indicating that the consumption of pork meat in Cyprus does not pose any health risk to children and adolescents. For doxycycline, a low ADI value of 3 μg/kg bw is established, and the detection frequency of this substance was 34.8% during 2017, where a maximum geometric mean concentration of 80.5 μg kg^−1^ was observed. The % EDI to ADI ratio for doxycycline is higher in relation to the rest of antimicrobials; however, it is still within the acceptance limits. 

## 4. Conclusions

In this work, a dietary exposure assessment was conducted on veterinary antibiotic residues found in raw pork meat among children and adolescents in Cyprus. The protocol of the study was based on a microbial screening method, followed by a LC-MS/MS confirmatory method [47]. The chemical analysis data were further used in a dietary exposure assessment model that was developed to estimate the daily intake of antibiotic residues through pork consumption in Cyprus for children and adolescents of both sexes. The results of this dietary exposure assessment indicate that, for all the substances, the percentage EDI to ADI ratio was less than 5.6, indicating no risk to human health. For doxycycline, the EDI to ADI ratio was higher than unit in most of the age groups. In general, the results suggest that pork meat does not present a potential health risk for children and adolescents aged 6–17 years old in Cyprus. However, the young consumers were more exposed to higher concentrations of antibiotic residues than adults. Children have higher body surface to body weight ratio and higher intake of calories and water compared to adults. Due to these differences in physiology compared to adults, children have higher exposure levels per kg bw, and they are more susceptible to various residues. Therefore, systematic exposure of antibiotics even in low concentrations, especially in early life, may have a negative impact in human health. That is why further investigation is strongly encouraged to protect human health, especially during developmental periods.

## Figures and Tables

**Figure 1 foods-09-01479-f001:**
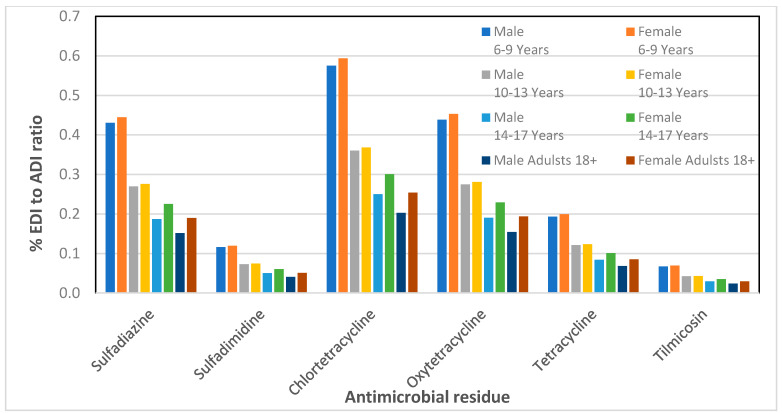
Percentage of estimated daily intake (EDI) to acceptable daily intake (ADI) ratio for six veterinary antibiotic residues found in pork meat for children, adolescents, and adults of the Cypriot population.

**Figure 2 foods-09-01479-f002:**
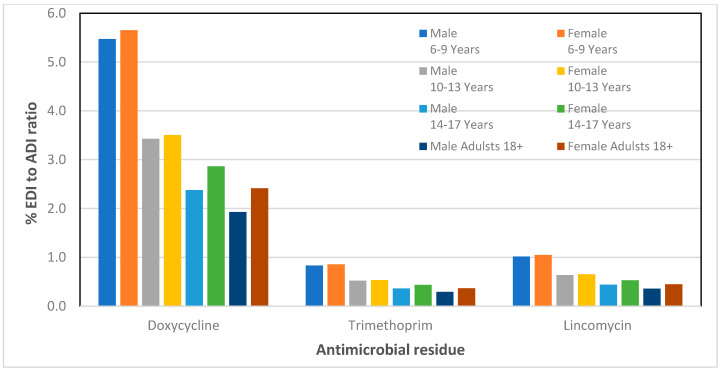
Percentage of estimated daily intake (EDI) to acceptable daily intake (ADI) ratio for doxycycline, trimethoprim, and lincomycin residues found in pork meat for children, adolescents, and adults of the Cypriot population.

**Table 1 foods-09-01479-t001:** Veterinary antibiotics and maximum residue limits (MRLs) in porcine muscle.

Antibiotic.	Antibiotic Class	MRL (μg·kg^−1^)
Dihydrostreptomycin	Aminoglycoside	500
Gentamycin	Aminoglycoside	50
Kanamycin	Aminoglycoside	100
Paromomycin	Aminoglycoside	500
Spectinomycin	Aminoglycoside	300
Streptomycin	Aminoglycoside	500
Cefalonium	Cephalosporin	N.E. ^i^
Cephapirin	Cephalosporin	50
Cefoperazone	Cephalosporin	N.E. ^i^
Ciprofloxacin	Fluoroquinolone	100
Danofloxacin	Fluoroquinolone	200
Enrofloxacin	Fluoroquinolone	100
Flumequine	Fluoroquinolone	200
Marbofloxacin	Fluoroquinolone	150
Norfloxacin	Fluoroquinolone	N.E. ^i^
Sarafloxacin	Fluoroquinolone	10
Nalidixic acid	Quinolone	N.E.^i^
Lincomycin	Lincosamide	100
Erythromycin	Macrolide	200
Spiramycin	Macrolide	200
Tilmicosin	Macrolide	N.E. ^i^
Ampicillin	Penicillin	50
Cloxacillin	Penicillin	300
Dicloxacillin	Penicillin	300
Penicillin-G (Benzylpenicilline)	Penicillin	50
Penicillin-V (Phenoxymethylpenicillin)	Penicillin	25
Nafcillin	β-lactam	N.E. ^i^
Oxacillin	β-lactam	300
Tiamulin	Pleuromutilin	100
Sulfadiazine	Sulphonamide	100
Sulfadimethoxine	Sulphonamide	100
Sulfadimidine	Sulphonamide	100
Sulfadoxine	Sulphonamide	100
Sulfamerazine	Sulphonamide	100
Sulfamethoxazole	Sulphonamide	100
Sulfamethoxypyridazine	Sulphonamide	100
Sulfanilamide	Sulphonamide	100
Sulfaguanidine	Sulphonamide	100
Sulfaquinoxaline	Sulphonamide	100
Sulfathiazole	Sulphonamide	100
Trimethoprim	Dehydroreductase inhibitor	50
Chlortetracycline	Tetracyclines	100
Doxycycline	Tetracyclines	100
Oxytetracycline	Tetracyclines	100
Tetracycline	Tetracyclines	100

^i^ Antibiotic with not established MRL level.

**Table 2 foods-09-01479-t002:** Estimated daily intake values in μg/kg body weight (bw)/day of antibiotics in male children and adolescents in Cyprus between 2012 and 2017.

	**2012**			**2013**			**2014**		
**Antimicrobial Residue**	**6–9 Years**	**10–13 Years**	**14–17 Years**	**6–9 Years**	**10–13 Years**	**14–17 Years**	**6–9 Years**	**10–13 Years**	**14–17 Years**
Sulfadiazine	0.09	0.05	0.04	0.02	0.01	0.01	0.07	0.04	0.03
Sulfadimidine	0.04	0.02	0.02	0.02	0.02	0.01	0.02	0.01	0.01
Sulfamethoxazole	0.10	0.06	0.04	<LOD ^i^	<LOD ^i^	<LOD ^i^	0.01	0.01	0.01
Trimethoprim	0.05	0.03	0.02	0.05	0.03	0.02	0.10	0.06	0.04
Lincomycin	<LOD ^i^	<LOD^i^	<LOD ^i^	0.31	0.19	0.13	0.10	0.06	0.04
Chlortetracycline	0.17	0.11	0.08	0.10	0.06	0.04	0.171	0.11	0.07
Doxycycline	0.08	0.05	0.04	0.09	0.05	0.04	<LOD ^i^	<LOD ^i^	<LOD ^i^
Oxytetracycline	0.11	0.07	0.05	0.05	0.03	0.02	0.13	0.08	0.06
Tetracycline	<LOD ^i^	<LOD^i^	<LOD ^i^	<LOD ^i^	<LOD ^i^	<LOD ^i^	0.06	0.04	0.02
Tilmicosin	0.02	0.01	0.01	<LOD ^i^	<LOD ^i^	<LOD ^i^	<LOD ^i^	<LOD ^i^	<LOD ^i^
	**2015**			**2016**			**2017**		
**Antimicrobial Residue**	**6–9 Years**	**10–13 Years**	**14–17 Years**	**6–9 Years**	**10–13 Years**	**14–17 Years**	**6–9 Years**	**10–13 Years**	**14–17 Years**
Sulfadiazine	0.01	0.01	0.01	0.01	0.01	0.01	0.06	0.04	0.03
Sulfadimidine	0.03	0.02	0.01	0.06	0.04	0.03	<LOD ^i^	<LOD ^i^	<LOD ^i^
Sulfamethoxazole	0.01	0.01	0.01	<LOD ^i^	<LOD^i^	<LOD ^i^	<LOD ^i^	<LOD ^i^	<LOD ^i^
Trimethoprim	0.05	0.03	0.02	<LOD ^i^	<LOD^i^	<LOD ^i^	0.04	0.03	0.02
Lincomycin	0.04	0.03	0.02	<LOD ^i^	<LOD^i^	<LOD ^i^	0.25	0.16	0.11
Chlortetracycline	0.09	0.06	0.04	<LOD ^i^	<LOD^i^	<LOD ^i^	<LOD ^i^	<LOD ^i^	<LOD ^i^
Doxycycline	<LOD ^i^	<LOD ^i^	<LOD ^i^	<LOD ^i^	<LOD^i^	<LOD ^i^	0.16	0.10	0.07
Oxytetracycline	0.07	0.04	0.03	<LOD ^i^	<LOD^i^	<LOD ^i^	0.05	0.03	0.02
Tetracycline	<LOD ^i^	<LOD ^i^	<LOD ^i^	<LOD ^i^	<LOD^i^	<LOD ^i^	<LOD ^i^	<LOD ^i^	<LOD ^i^
Tilmicosin	0.03	0.02	0.01	0.01	0.01	0.01	0.02	0.01	0.01

^i^ LOD: limit of detection.

**Table 3 foods-09-01479-t003:** Estimated daily intake values in μg/kg bw/day of antibiotics in female young population in Cyprus between 2012 and 2017.

	**2012**			**2013**			**2014**		
**Antimicrobial Residue**	**6–9 Years**	**10–13 Years**	**14–17 Years**	**6–9 Years**	**10–13 Years**	**14–17 Years**	**6–9 Years**	**10–13 Years**	**14–17 Years**
Sulfadiazine	0.09	0.06	0.04	0.02	0.02	0.01	0.07	0.04	0.04
Sulfadimidine	0.04	0.02	0.02	0.02	0.02	0.01	0.02	0.01	0.01
Sulfamethoxazole	0.11	0.07	0.05	<LOD ^i^	<LOD ^i^	<LOD ^i^	0.01	0.01	0.01
Trimethoprim	0.05	0.03	0.03	0.05	0.03	0.02	0.11	0.07	0.05
Lincomycin	<LOD ^i^	<LOD ^i^	<LOD ^i^	0.32	0.19	0.16	0.10	0.06	0.05
Chlortetracycline	0.18	0.11	0.09	0.10	0.06	0.05	0.18	0.11	0.09
Doxycycline	0.08	0.05	0.04	0.09	0.06	0.04	<LOD ^i^	<LOD ^i^	<LOD ^i^
Oxytetracycline	0.11	0.07	0.06	0.05	0.03	0.03	0.14	0.08	0.07
Tetracycline	<LOD ^i^	<LOD ^i^	<LOD ^i^	<LOD ^i^	<LOD ^i^	<LOD ^i^	0.06	0.04	0.03
Tilmicosin	0.02	0.01	0.01	<LOD ^i^	<LOD ^i^	<LOD ^i^	<LOD ^i^	<LOD ^i^	<LOD ^i^
	**2015**			**2016**			**2017**		
**Antimicrobial Residue**	**6–9 Years**	**10–13 Years**	**14–17 Years**	**6–9 Years**	**10–13 Years**	**14–17 Years**	**6–9 Years**	**10–13 Years**	**14–17 Years**
Sulfadiazine	0.01	0.01	0.01	0.01	0.01	0.01	0.07	0.04	0.03
Sulfadimidine	0.03	0.02	0.02	0.06	0.04	0.03	<LOD ^i^	<LOD ^i^	<LOD ^i^
Sulfamethoxazole	0.01	0.01	0.01	<LOD ^i^	<LOD ^i^	<LOD ^i^	<LOD ^i^	<LOD ^i^	<LOD ^i^
Trimethoprim	0.05	0.03	0.02	<LOD ^i^	<LOD ^i^	<LOD ^i^	0.04	0.03	0.02
Lincomycin	0.04	0.03	0.02	<LOD ^i^	<LOD ^i^	<LOD ^i^	0.26	0.16	0.13
Chlortetracycline	0.09	0.06	0.04	<LOD ^i^	<LOD ^i^	<LOD ^i^	<LOD ^i^	<LOD ^i^	<LOD ^i^
Doxycycline	<LOD ^i^	<LOD ^i^	<LOD ^i^	<LOD ^i^	<LOD ^i^	<LOD ^i^	0.17	0.11	0.09
Oxytetracycline	0.07	0.04	0.04	<LOD ^i^	<LOD ^i^	<LOD ^i^	0.05	0.03	0.02
Tetracycline	<LOD ^i^	<LOD ^i^	<LOD ^i^	<LOD ^i^	<LOD ^i^	<LOD ^i^	<LOD ^i^	<LOD ^i^	<LOD ^i^
Tilmicosin	0.03	0.02	0.01	0.01	0.01	0.01	0.02	0.01	0.01

^i^ LOD: limit of detection.

**Table 4 foods-09-01479-t004:** Estimated daily intake values (EDI) and percentage of EDI to acceptable daily intake (ADI) ratio for male children and adolescents using the maximum geometric mean concentration.

	EDI (μg/kg bw/day)		%EDI to ADI Ratio ^iii^						
Antimicrobial/ADI (μg/kg bw)	Year ^i^	C ^ii^ (μg/kg)	6–9 Years	10–13 Years	14–17 Years	6–9 Years	10–13 Years	14–17 Years	Adults 18+
Sulfadiazine/0–20	2012	42.2	0.09	0.05	0.04	0.43	0.27	0.19	0.15
Sulfadimidine/0–50	2016	28.4	0.06	0.04	0.02	0.12	0.07	0.05	0.04
Sulfamethoxazole ^iv^	2012	50.3	0.10	0.06	0.05	-	-	-	-
Trimethoprim/12.5	2014	50.9	0.10	0.06	0.05	0.83	0.52	0.36	0.29
Lincomycin/0–30	2013	149.4	0.31	0.19	0.13	1.02	0.64	0.44	0.36
Chlortetracycline/0–30	2012	84.6	0.17	0.11	0.08	0.58	0.36	0.25	0.20
Doxycycline/0–3	2017	80.5	0.16	0.10	0.07	5.47	3.43	2.38	1.93
Oxytetracycline/0–30	2014	64.5	0.13	0.08	0.06	0.44	0.27	0.19	0.15
Tetracycline/0–30	2014	28.4	0.06	0.04	0.02	0.19	0.12	0.08	0.07
Tilmicosin/0–40	2015	13.2	0.03	0.02	0.01	0.07	0.04	0.03	0.02

^i^ Year of the maximum concentration for the antimicrobials; ^ii^ Geometric mean of the concentrations of antimicrobials; ^iii^ % EDI to ADI ratio of less than 100% means that the estimated intake is lower than the acceptable intake; ^iv^ For Sulfamethoxazole ADI value is not defined.

**Table 5 foods-09-01479-t005:** Estimated daily intake values (EDI) and percentage of EDI to acceptable daily intake (ADI) ratio for female children and adolescents using the maximum geometric mean concentration.

			EDI (μg/ kg bw/ day)			%EDI to ADI Ratio ^iii^			
Antimicrobial/ADI (μg/kg bw)	Year ^i^	C ^ii^ (μg/kg)	6–9 Year	10–13 Year	14–17 Year	6–9 Year	10–13 Year	14–17 Year	Adults 18+
Sulfadiazine/0–20	2012	42.2	0.09	0.06	0.04	0.44	0.28	0.23	0.19
Sulfadimidine/0–50	2016	28.4	0.06	0.04	0.03	0.12	0.07	0.06	0.05
Sulfamethoxazole ^iv^	2012	50.3	0.11	0.07	0.05	-	-	-	-
Trimethoprim/12.5	2014	50.9	0.11	0.07	0.05	0.86	0.53	0.43	0.37
Lincomycin/0–30	2013	149.4	0.32	0.19	0.16	1.05	0.65	0.53	0.45
Chlortetracycline/0–30	2012	84.6	0.18	0.11	0.09	0.59	0.37	0.30	0.25
Doxycycline/0–3	2017	80.5	0.17	0.11	0.09	5.65	3.51	2.86	2.41
Oxytetracycline/0–30	2014	64.5	0.14	0.08	0.07	0.45	0.28	0.23	0.19
Tetracycline/0–30	2014	28.4	0.06	0.04	0.03	0.20	0.12	0.10	0.09
Tilmicosin/0–40	2015	13.2	0.03	0.02	0.01	0.07	0.04	0.04	0.03

^i^ Year of the maximum concentration for the antimicrobials; ^ii^ Geometric mean of the concentrations of antimicrobials; ^iii^ % EDI to ADI ratio of less than 100% means that the estimated intake is lower than the acceptable intake; ^iv^ For Sulfamethoxazole ADI value is not defined.

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
