# Peer review of "Dietary Exposure Assessment of Veterinary Antibiotics in Pork Meat on Children and Adolescents in Cyprus"

_foods, 2020, doi:10.3390/foods9101479_

Round 1

Reviewer 1 Report

The authors addressed correctly the mistakes highlighted.

Author Response

Thank you for the positive comment

Reviewer 2 Report

The manuscript provided a dietary exposure assessment of veterinary antibiotics in pork meat for Cypriots. While such assessment is of great importance to public health, this manuscript is too brief and descriptive. The manuscript is covered with tables and figures with very little discussion.

Specific comments:

  • Lines 34-35, page 2: the authors cited a reference here without a clear reason. What were the messages the authors would like to convey through this study?
  • Lines 14-18, page 3: the possibility of negative health impact by low concentration antibiotics exposure during early life is more of a toxicological question. The authors should explain how this study could address that question and discuss it further in the discussion part.
  • Lines 34-35, page 3: where were the internal standards from?
  • Section 3.1, page 6: the author should revise this paragraph the reflect how the data was chosen for the exposure assessment rather than summarize a published study.
  • Lines 28-29, page 6: while a product with antibiotics concentrations higher than their corresponding MRLs is banned, there is no guarantee such products are not available on the market. In the lines 39-40, page 2, the authors also stated that higher than MRLs concentrations have been reported. Therefore the authors should still include these results in the study, for example under a worst-case scenario.
  • Table 2 and other tables: too many significant figures have been used.
  • Section 3.3: the most important impact of this manuscript is the exposure assessment. Therefore, the authors should dive much more in-depth into this matter. For example, the authors could investigate the temporal trend of the estimated intake in different age/gender groups combination.

Author Response

Specific comments:

  1. Lines 34-35, page 2: the authors cited a reference here without a clear reason. What were the messages the authors would like to convey through this study?

RESPONSE: Reference 37 was cited based on the suggestions of the Rev1 in the previous revision of the manuscript (review report 1, comment 4: Please refer to following references regarding exposure assessment. Choi, S. Y., ji Kwon, N., Kang, H. S., Kim, J., Cho, B. H., & Oh, J. H. (2020). Residues determination and dietary exposure to ethoxyquin and ethoxyquin dimer in farmed aquatic animals in South Korea. Food Control, 111, 107067). Lines 34-35 have been then added to highlight the content of this publication.

  1. Lines 14-18, page 3: the possibility of negative health impact by low concentration antibiotics exposure during early life is more of a toxicological question. The authors should explain how this study could address that question and discuss it further in the discussion part.

RESPONSE:  This is not a risk assessment study; this is an exposure assessment study that could be used at the later steps of a risk assessment procedure. 

The following are clearly mentioned in the introduction section : “we thought that it would be of interest to contact a dietary exposure assessment that could be used as the preliminary step of a risk assessment procedure over the consumption of pork meat among children and adolescents in Cypriot population. When including a dietary exposure assessment, the risk assessment provides the scientific basis for the establishment of appropriate regulations and guidelines on the use of veterinary antibiotics in food producing animals. This will ensure that safety requirements will be protective for the consumers and appropriate for use in national and international scale”.

  1. Lines 34-35, page 3: where were the internal standards from?

RESPONSE: ISTDs were purchased from Sigma Aldrich and this is now mentioned in the manuscript.

Section 3.1, page 6: the author should revise this paragraph the reflect how the data was chosen for the exposure assessment rather than summarize a published study.

RESPONSE: We have summarized only the analytical LC-MS/MS method, results are presented clearly in this study, we have slightly modified the specific sentence to clearly demonstrate this.

  1. Lines 28-29, page 6: while a product with antibiotics concentrations higher than their corresponding MRLs is banned, there is no guarantee such products are not available on the market. In the lines 39-40, page 2, the authors also stated that higher than MRLs concentrations have been reported. Therefore the authors should still include these results in the study, for example under a worst-case scenario.

RESPONSE: It is now clearly mentioned in the manuscript that: “In particular, the Veterinary Services of the Ministry of Agriculture, Rural Development and Environment in Cyprus have the legal power to take appropriate measures in cases of detected residues violations, restriction or prohibition of the placing on the market; monitoring, recall, withdrawal and/or destruction. A national program for testing of slaughtered pigs under which every animal is tested for the presence of antimicrobial residues. Animals are detained pending a result, if non-compliant, the animals are destroyed and in no case possible does the products reach the Cyprus market.”

This information can be verified under the Directorate’s-General for Health and Food Safety of the European Commission report (page 48), based on information for control systems for food and feed safety, animal health, animal welfare, plant health and quality labelling, in Cyprus (DG(SANTE)/2017/6037 Final. Version February 2018).

Therefore, products of animal origin with antibiotics concentrations higher than their corresponding MRLs is banned from the Cypriot market, and there is no point to include these results in the current study.

  1. Table 2 and other tables: too many significant figures have been used.

RESPONSE: in tables 2 and 3 all numbers have been rounded to 3 decimals.

  1. Section 3.3: the most important impact of this manuscript is the exposure assessment. Therefore, the authors should dive much more in-depth into this matter. For example, the authors could investigate the temporal trend of the estimated intake in different age/gender groups combination.

RESPONSE:

Thank you for your comment. To investigate the temporal trend of the estimated daily intake in different age/gender groups combination, specific models should be developed; this comment trigger some views for conducting research in this area in the future. However, we do believe that too much information has already been presented in the current manuscript.

Reviewer 3 Report

FOODS 935749
The manuscript entitled: “Dietary exposure assessment of veterinary antibiotics in pork meat on children and adolescent in Cyprus”, focus on the importance of the evaluation the exposure to antibiotics present in pork in the Cypriot juvenile population.
The paper fits in the scope of the journal, the title is appropriate, informative, concise, and clear. The abstract comprehensive, containing the essential information of the article. The structure of the article is well organized.
The manuscript is scientifically interesting and could be useful for further studies. The tables are exhaustive and clear. The discussion and conclusion are well written and the study supported by appropriate evidence. Finally, the English used in the article readable and good enough to convey the scientific meaning.

Author Response

Thank you for this positive opinion on the manuscript.

Round 2

Reviewer 2 Report

The manuscript in its current form is still very descriptive and does not warrant a high-quality publication.

Author Response

Reviewer comment:The manuscript in its current form is still very descriptive and does not warrant a high-quality publication.

Response: Thank you for the revision this manuscript.